

# Climatology of migrating and non-migrating tides observed by three meteor radars in the southern equatorial region

Jianyuan Wang[1,2], Wen Yi[1,2,3], Jianfei Wu[1,2,4], Tingdi Chen[1,2,4], Xianghui Xue[1,2,4,5],

Robert. A. Vincent[6], Iain. M. Reid[6,7], Paulo P. Batista[8], Ricardo A. Buriti[9], Toshitaka

Tsuda[10], Xiankang Dou[1,11]

[1]CAS Key Laboratory of Geospace Environment, Department of Geophysics and Planetary Sciences,

University of Science and Technology of China, Hefei, China

[2]Mengcheng National Geophysical Observatory, School of Earth and Space Sciences, University of

Science and Technology of China, Hefei, China

[3]State Key Laboratory of Space Weather, Chinese Academy of Sciences, Beijing, China

[4]CAS Center for Excellence in Comparative Planetology, Hefei, China

[5]Synergetic Innovation Center of Quantum Information and Quantum Physics, University of Science and

Technology of China, Hefei, China

[6]School of Physical Sciences, University of Adelaide, Adelaide, South Australia, Australia

[7]ATRAD Pty Ltd, Thebarton, Australia

[8]National Institute for Space Research, S. J. dos Campos, 12227-010, Brazil

[9]Federal University of Campina Grande, PB, Brazil

[10]Research Institute for Sustainable Humanosphere (RISH), Kyoto University, Uji, Kyoto, Japan

[11]Wuhan University, Wuhan, China

*Correspondence to*: Xianghui Xue (xuexh@ustc.edu.cn)



**Abstract.** We present a study of migrating and non-migrating tidal winds observed simultaneously by three meteor radars situated in the southern equatorial region. The radars are located at Cariri (7.4° S, 36.5° W), Brazil, Kototabang (0.2 °S, 100.3 °E), Indonesia and Darwin (12.3 °S, 130.8 °E), Australia. Harmonic analysis was used to obtain amplitudes and phases for diurnal and semidiurnal solar migrating and non-migrating tides between 80 and 100 km altitude during the period 2005 to 2008. They include the important tidal components of diurnal westward-propagating zonal wavenumber 1 (DW1), diurnal eastward-propagating zonal wavenumber 3 (DE3), semidiurnal westward-propagating zonal wavenumber 2 (SW2), and semidiurnal eastward-propagating zonal wavenumber 2 (SE2). In addition, we also present a climatology of these wind tides and analyze the reliability of the fitting through the reference to Whole Atmosphere Community Climate Model (WACCM) winds. The analysis suggests that the migrating tides could be well fitted by the three different radars, but the non-migrating tides might be overestimated. The results based on observations were also compared with the Climatological Tidal Model of the Thermosphere (CTMT). In general, climatic features between observations and model migrating tides were satisfactory in both wind components. However, the features of the DW1, DE3 and SW2 amplitudes in both wind components were slightly different from the results of the CTMT models. This result is probably because tides could be enhanced by the 2006 northern hemisphere stratospheric sudden warming (NH-SSW) event.

## 1. Introduction

Dynamical processes in the upper mesosphere and lower thermosphere (MLT) region provide a significant key to understanding coupling processes between the lower atmosphere and the thermosphere/ionosphere system. On one hand, the dynamics in the MLT region are strongly forced by atmospheric oscillations (such as gravity waves, tides, and planetary waves) originating in the lower atmosphere and are therefore dynamically coupled to the troposphere and stratosphere. On the other hand, the MLT region represents the lower boundary of the thermosphere/ionosphere (TI) system, and knowledge of these processes is necessary for studies of the thermosphere and/or ionosphere with regard to space weather. Among them, atmospheric tides are significant oscillations that can dominate the dynamics in the MLT region.

Atmospheric tides are global-scale atmospheric oscillations with periods that are harmonics of a solar day. These tides are predominantly generated in the troposphere and stratosphere and then propagate vertically to the MLT, where they reach large amplitudes. These atmospheric tides can be classified as migrating tides and non-migrating tides. Migrating tides propagate westward with the apparent motion of the



sun, and non-migrating tides can propagate either westward or eastward or remain stationary. Migrating diurnal (24-hour period, westward propagating, zonal wavenumber 1) and semidiurnal (12-hour period, westward propagating, zonal wavenumber 2) tides are the dominant modes that have been extensively studied. In general, 24-hour tides (diurnal tides) are dominant at low latitudes, whereas 12-hour tides (semidiurnal tides) have larger amplitudes at middle-to-high latitudes, peaking at ~60° latitude (Hagan et al., 1999). A growing amount of evidence shows that non-migrating tides, whose phase speeds do not follow the apparent motion of the sun, can also achieve large amplitudes in the MLT region and contribute significantly to the variability in this region (Forbes et al., 2008; Oberheide et al., 2011).

Space-based instruments provide a global picture of the MLT wind regime. Wu et al. (2008) used the TIMED Doppler Interferometer (TIDI) zonal and meridional winds to fit migrating and non-migrating tides. However, these instruments usually cannot provide good time resolution and potentially bias temporal and spectral information.

MLT winds can be observed by radio radar systems, meteor radar (MR) or medium frequency (MF) radar. Since there are potential biases in different kinds of radar observations (Jacobi et al., 2009), wind measurements should be performed with the same or at least similar setups. Recently, Kleinknecht et al. (2014) presented results on quasi-stationary planetary waves 1 and 2 based on the Super Darn radar network, which mainly occupies western longitudes. He et al. (2018) also demonstrated the climatology of these oscillations for an average height.

Therefore, in this study, we make use of the horizontal mesospheric winds observed by three meteor radars near the equator, all being MRs, each with a similar layout and instrumentation and to which we apply the same analytical procedures and compare these results with winds from models. We analyze the parameters of the dominant migrating and non-migrating tides, discuss the feasibility of tidal fits by using horizontal winds from the Whole Atmosphere Community Climate Model (WACCM) dataset, and compare the similarities and differences in the climatologies of these tidal components between MR observations and the Climatological Tidal Model of the



Thermosphere (CTMT).

**2. Data and methods**

In this study, horizontal wind data from three MRs located at Cariri (7.4° S, 36.5° W),
Kototabang (0.2° S, 100.3° E) and Darwin (12.3° S, 130.8° E), hereinafter referred to

as CMR, KMR and DMR, respectively, were used. Table 1 summarizes the operating
frequencies, geographic coordinates and observational time spans for the MR dataset
used in the current study. Figure 1 shows the geographic locations of these three meteor
radars.    CMR is running by the National Institute for Space Research (INPE), Brazil,
DMR is running by the University of Adelaide, and KMR is a part of the CPEA project,

supported by the Research Institute for Sustainable Humanosphere (RISH), Kyoto
University.

| Meteor radar | Geographic locations | Frequency | Data used in this study |
|---|---|---|---|
| Cariri (CMR) | 7.4° S, 36.5° W | 35.2 MHz | 1/1/2005-12/31/2008 |
| Kototabang (KMR) | 0.2° S, 100.3° E | 37.7 MHz | 11/1/2002-9/30/2017 |
| Darwin (DMR) | 12.3° S, 130.8° E | 47.5 MHz | 1/1/2005-12/31/2008 |

**Table 1.** Geographic locations, working frequencies and observational time periods of
the MRs used in this study.

The radar system transmitted the radio wave and coherently detected the meteor trail

reflections. The range is obtained from the time interval between transmission and
detection, and the azimuth and elevation angles are calculated from the phase
differences between antennas. Assuming that the vertical wind is negligible, the
horizontal wind can be determined by the Doppler shift following the algorithms used
in Hocking et al. (2001) and Holdsworth et al., 2004.

The KMR and DMR are used to measure horizontal winds from 80 to 100 km. Temporal
and altitudinal resolutions are 1 hour and 2 km, respectively; the temporal resolution of
CMR is also 1 hour, while the altitude range of the CMR wind is available from 81 to
99 km with a resolution of 3 km. To use these data for fitting migrating tides and non-
migrating tides, the vertical resolution of horizontal wind data obtained by CMR was



linearly interpolated to 2 km within the altitude range from 82 to 98 km.

## 2.1. Decomposition approach for tides

Figure 2 (a) to (f) shows the hourly mean zonal and meridional winds averaged during 2005 to 2008 for the three stations. Although the wind velocities differ slightly among the three stations, the wind patterns within the 80 to 100 km altitude range are quite similar. The dominant feature is a 24-hour diurnal oscillation in both wind components across the three MR stations. The largest anomalies of the wind velocities are approximately 60 m/s in the meridional component, and the anomalies of the zonal wind speeds are slightly weaker at approximately 20 m/s. At all stations, the downward phase propagation indicates upward wave propagation from the lower atmosphere. The phase shifts by approximately 16 hours within the 20 km vertical range, indicating vertical wavelengths of approximately 30 km.

Moreover, by comparing the meridional wind patterns in Figure 2 (b), (d) and (f), one finds an evident time delay of approximately 8 hours between the DMR and KMR and approximately 12 hours between the DMR and CMR. Given that the 24-hour oscillation propagates westward, Darwin should be located west of Kototabang; however, Darwin is located at 130.8°E and Kototabang is located at 100.3°E, revealing the complex tidal composition near the equator and that the strongest tidal component propagates eastward at these three locations.

As shown in Figure 2 (g) to (l), the 24-hour oscillation is the most significant in the Lomb-Scargle (LS) periodogram (Lomb, 1976; Scargle, 1982) of the hourly zonal and meridional winds over the three stations, and the 12-hour oscillation is the second-largest signal. In addition, other periods could also be found in the LS periodogram; for example, the quasi-two-day oscillation is clearly evident in the meridional wind.

In principle, the series of hourly zonal and meridional winds in the individual altitude bins from three stations could be used to calculate the amplitudes and phases of various tidal components, including diurnal, semidiurnal and terdiurnal tides with zonal wavenumbers ranging between eastward-propagating 3 and westward-propagating 3. However, at least six stations with different longitudes are required to accurately





calculate the oscillations with zonal wavenumbers of three at most. Here, we assume that in comparison with those significant waves, other potential waves are negligible. In order to accurately evaluate the dominant tidal components in the MLT wind and discuss the accuracy of the fitting method, it is necessary to use a high-resolution MLT

wind model, and for WACCM, the neutral winds in MLT region have high temporal and spatial resolution and satisfactory accuracy. Therefore, the WACCM is used to fit each tidal component to determine which waves are dominant near the latitudes of CMR, KMR, and DMR.

## 2.2. Determine dominant tides

WACCM is a high-top model based on the Community Earth System Model (version 1) framework (Hurrell et al., 2006), which is an optional variation of the Community Atmosphere Model version 4 (CAM4) that has been designed to investigate the interactions among chemistry, radiation, and dynamics and their impact on the Earth's climate system (Neale et al., 2013). The current study uses the specific dynamics (SD)

version of WACCM, in which neutral temperatures and winds below 50 km are nudged toward NASA's Modern-Era Retrospective analysis for Research and Applications (MERRA) (Lamarque et al., 2012). The present integrations have a horizontal resolution of 2.5° in longitude and 1.9° in latitude. There are 144 vertical levels; the vertical resolution is approximately 0.5 km ranging from 80 to 100 km. The time step

is 30 minutes.

Least-square fits are performed on the wind data for each 3-day window throughout the series, and only days with data that are available for more than 18 hours each day are used. Because the WACCM zonal and meridional wind data are available on a global scale, the zonal wavenumber is taken from eastward-propagating 7 to westward-

propagating 7 for fitting all tidal components. In addition, it should be noted that quasi-two-day oscillations may also contribute to neutral winds in the 3-day window, so these quasi-two-day components with all zonal wavenumbers should be added to the tidal fits. Following the above method, the meridional or zonal wind variations in time and longitude at a given altitude can be written as follows:





$$u(z, \theta, t) = u_0(z) + \sum_{n=1}^{3} \sum_{s=-7}^{7} c_{n,s}(z) \cos\left(\frac{2\pi n}{T}t + \frac{2\pi s}{2\pi}\lambda + \varphi_{n,s}(z)\right) +$$

$$\sum_{s=-7}^{7} c_{4,s}(z) \cos\left(\frac{2\pi}{2T}t + \frac{2\pi s}{2\pi}\lambda + \varphi_{4,s}(z)\right), \qquad (1)$$

where u means either the zonal or meridional wind; z means altitude; $\theta$ means latitude; t means time; $\lambda$ means longitude (rad); n means temporal wavenumber and s means zonal wavenumber; T equals 24 hours; $\varphi$ means phase; $u_0$ means zonal mean zonal wind; the second section means tidal components comprising 24-, 12- and 8-hour oscillations; and the third section means quasi-two-day oscillation components.

Figure 3 presents the tidal amplitudes averaged at all times and all altitudes fitted by the WACCM wind dataset at 6.63 °S, which is the average latitude of these three MRs, for each tidal component. The most significant tidal components at 6.63 °S are migrating tides, including diurnal westward-propagating zonal wavenumber 1 (DW1) and semidiurnal westward-propagating zonal wavenumber 2 (SW2). In addition, diurnal eastward-propagating zonal wavenumber 3 (DE3) and semidiurnal eastward-propagating zonal wavenumber 2 (SE2) are also stronger at 6.63 °S.

Because MR data only provide three different longitudes, the fits of the tides with zonal wavenumbers greater than or equal to two cannot be considered. Therefore, we only decompose DW1, SW2, DE3 and SE2 into these four tidal components and quasi-two-day oscillations with zonal wavenumbers from eastward-propagating 3 to westward-propagating 3 when fitting tides with CMR, KMR and DMR.

## 3. Results

Figure 4 presents the monthly mean amplitudes of diurnal and semidiurnal tides from 2005 to 2008 observed in the altitude range of 80 to 100 km at the three stations, and the observations for Cariri are reported by Buriti et al. (2019). The figure shows that both diurnal and semidiurnal tides have larger amplitudes in the meridional component than in the zonal component. For diurnal tides, the amplitudes in the meridional component are between ~5 and 65 m/s, while those in the zonal component are between ~5 and 40 m/s; for semidiurnal tides, the amplitudes in the meridional component are between ~5 and 40 m/s, and those in the zonal component are between ~5 and 25 m/s. Figure 4 (a) to (f) shows that the diurnal tidal amplitudes in both components reach their



largest values in February/September and reach minimum values in June. This feature is slightly different at Kototabang: the peak amplitudes occur in December/January and July-September, and the amplitudes are stronger during January to March, while the minima occur in April. In most cases, the amplitudes increase with increasing altitude, but the amplitudes at Kototabang in the meridional component in the range from 84 to 88 km are smaller than those at lower altitudes during August. At Darwin, the meridional diurnal component has large amplitudes in February and September and these are stronger than those at Cariri and Kototabang. This feature in February is observed in both wind components and can be captured throughout almost the entire vertical region being considered.

Figure 4 (g) to (l) shows that the semidiurnal tidal amplitudes in the meridional component are stronger in February/May-June/September-November, and in particular, those amplitudes at Kototabang are smaller in October to December than at the other stations; the semidiurnal tides in the zonal component are so weak that there are no significant seasonal variations, showing only that the amplitudes become stronger with increasing altitude.

By using the winds over a single meteor radar, it is easy to calculate the diurnal and semidiurnal tides, but the migrating and non-migrating tides cannot be separated, and the role of each tidal component cannot be clearly determined. To extract the migrating and non-migrating tides, the three meteor radars have been jointly analyzed by function (1) to calculate amplitudes and phases for all dominant tidal components.

To show the features of migrating tides, Figure 5 shows the monthly mean amplitudes of diurnal and semidiurnal migrating tides in composite years at altitudes ranging from 82 to 98 km. The diurnal migrating tides have amplitudes between ~1 and 12 m/s in both components, and the semidiurnal tidal amplitudes reach the same maximum of 12 m/s in the zonal component, while the maximum of these amplitudes in the meridional component is stronger at approximately 17 m/s. The diurnal migrating tides in both components exhibit almost the same pattern, with amplitude maxima in March and September and minima in June. For the semidiurnal migrating tides, the amplitudes



peaked in February/March and in the region above 94 km in June, showing similar seasonal variations in migrating tidal amplitudes in both components.

Figure 6 shows the monthly mean amplitudes of the DE3 and SE2 tides in the composite year in the altitude range from 82 to 98 km. The DE3 tide, the most significant diurnal non-migrating tidal component, has amplitudes between ~1 and 40 m/s in the meridional components and between ~1 and 25 m/s in the zonal component; the SE2 tidal amplitudes reach a maximum of 17 m/s in the meridional component, while the maximum amplitudes in the zonal component are weaker by ~5 m/s. The DE3 tides in both components exhibit almost the same pattern, with amplitude maxima in February/March and September and minima in June; specifically, the DE3 tidal amplitudes in the meridional component also reach a maximum in December. For the SE2 tides, the amplitudes in the meridional component peaked in February/March and September and in the region above 94 km in June, but the amplitudes in the zonal component had 4 peak values in January to March, June, August/September and November, showing different seasonal variations in non-migrating tidal amplitudes between each component.

Comparing the amplitudes of migrating tides and non-migrating tides, the largest amplitude in DW1 is approximately 12 m/s, which is one-third of that in DE3; the largest amplitude in SW2 is approximately 17 m/s, which is similar to that in SE2.

As shown in Figure 7, all four tidal component phases (Universal Time of maximum at the longitude of 0° in hours) are observed earlier in the zonal component than in the meridional component at most altitudes, and these phases are approximately symmetrical around July at all altitudes considered. The phase differences between zonal and meridional components, shown in Figure 7(i) to (l) are mostly equal to -6 and -3 hours (equivalent to -90°) for the diurnal and semidiurnal tides, which are consistent with previous results(Oberheide et al., 2006; G. Liu et al., 2020). In addition, the diurnal tidal phase changes with altitude by approximately 30 hours are slightly larger than the semidiurnal tidal phase changes, which are approximately 12 hours.

## 4. Discussion



Our work has focused on using three MRs with similar latitudes to fit migrating and non-migrating tides, and we have also presented the local tides fitted by these three MRs. In past studies, some works have used multiple MRs with similar latitudes to fit migrating tides, but these studies are mainly focused on diurnal migrating tides, and there are only a few works related to semidiurnal migrating tides. For example, He et al. (2019) extracted high-order solar migrating tides through three MRs at boreal mid-latitudes, and few works have used three or more MRs during the same time span to decompose non-migrating tidal components. This finding may be because there are very few MRs near similar latitudes and no more than three locations, such as Mohe (54 °N, 123 °E), Juliusruh (55 °N, 12 °E), and Kazan (55.7 °N, 49 °E) near the mid-latitudes of the Northern Hemisphere and Cariri (7.4 °S, 36.5 °W), Kototabang (0.2 °S, 100.3 °E), and Darwin (12.3 °S, 130.8 °E) in the southern equatorial region, as used in this work. Since the MR winds we use to decompose the non-migrating tides have only three different longitudes, an analysis of the reliability of the fitting is necessary. The WACCM model is used to discuss this aspect.

We use the wind data of the WACCM model to fit tides in altitudes ranging from 80 to 100 km at 6.63 ° S with whole longitudes, in which the diurnal and semidiurnal tides are extracted using -7 to 7 zonal wavenumbers. Then, we use the horizontal winds at the three locations of Cariri, Kototabang, and Darwin to decompose the tidal components of DW1, SW2, DE3 and SE2. Figure 8 compares the tidal amplitudes at 86 km fitted by the whole longitudes with the three-point fitting results and calculates the correlation between the two results for each tidal component. For migrating tides, the seasonal features of the full-latitude circle fitting and the three-point fitting are very similar, and their correlations are satisfactory; both correlations are higher than 0.8. For the meridional components, the correlation coefficients are already higher than 0.9. For non-migrating tides, the main seasonal features can be reproduced well, but the correlation between them is lower. In addition, it is noted that the three-point fitting will lead to a certain underestimation of migrating tides and a certain overestimation of DE3, especially SE2, for non-migrating tides; other non-migrating tidal components are small





and will not be discussed here. Clark et al. (2002) mentioned in their work that when the winds at two locations are used to fit the tides, the longitudinal difference between the two locations should be less than half the wavelength of this tidal component. In our work, we use three-point fitting to extract tides. For the main semidiurnal non-

5 migrating tide SE2, which is near the equator, the longitudinal difference between Darwin and Kototabang is ~30°, which is less than 90° for half the wavelength; the longitudinal difference between Cariri and Darwin is ~190° larger than half the wavelength. Cariri and Kototabang differ by ~136°, also exceeds half the wavelength.

For the main diurnal non-migrating tide, DE3, which is near the equator, the

10 longitudinal difference between Darwin and Kototabng is less than 60° of a half-wavelength; the longitudinal difference between Cariri and Darwin is ~190°, and the difference between Cariri and Kototabang is 136°, both exceeding half-wavelength. For these three-point fits, the tides with westward- or eastward-propagating zonal wavenumbers 2 and 3 could be partly fitted by this longitudinal distribution, but the

15 fitted tides between Cariri and Darwin and Cariri and Kototabang may be less accurate if the interval exceeds half the wavelength. Nevertheless, we also find that in the tidal fitting based on the WACCM model, the three-point fitting tidal amplitudes are more satisfactory in fitting the migrating tides with zonal wavenumbers 1 and 2, and they are very similar to the result of the tidal fitting with the full-latitude circle; although the

20 fitting of non-migrating tides of zonal wavenumbers 2 and 3 has a poor correlation, the main seasonal variations can be reflected, which indicates that it is feasible to fit the migrating and non-migrating tides based on three-point winds.

For further discussion, we compared the observation results with the Climatological Tidal Model of the Thermosphere (CTMT), which was proposed and completed by J.

Oberheide et al. (2011). All data regarding the diurnal and semidiurnal components were downloaded from the Atmospheric & Space Physics at Clemson University, available at http://globaldynamics.sites.clemson.edu/articles/ctmt.html. This software is a 2-dimensional model, which is based on tidal observations made in the MLT region that are extended into the thermosphere using Hough mode extension (HME) modeling



as a function of latitude (from 90° S to 90° N), altitude (from 80 km to 400 km), and month (from January to December). The observation data comprise the TIMED Doppler Interferometer (TIDI) zonal and meridional wind tides, Sounding the Atmosphere using Broadband Emission Radiometry (SABER) temperature tides and the Challenging Minisatellite Payload (CHAMP) neutral density tides in the 2002 to 2008 composite years. CTMT provides the 13 most significant diurnal and semidiurnal tidal components for temperature, density, and zonal, meridional and vertical winds, including DW2, DW1, D0, DE1, DE2, DE3, SW4, SW3, SW1, S0, SE1, SE2, and SE3. In addition, CTMT is only valid for 110 solar flux unit (sfu) due to the tidal dissipation sensitivity to thermospheric temperature.

The CTMT results also support that the dominant tidal components near 5°S are DW1, SW2, DE3 and SE2, as shown in Figure 3, so this conclusion will not be repeated. Figure 9 presents the DW1, DE3, SW2, and SE2 tidal amplitudes in both wind components in the altitude range from 80 to 100 km based on CTMT and MR observations. In the right column, the CTMT results show that the diurnal migrating tides, DW1, reach the largest value in March and have the second strongest amplitudes in October in both wind components. Notably, the DW1 tidal amplitudes in the meridional components are larger than those in the zonal components. These climatological features are similar to our fitting results by the three MRs in the left column, and the maximum of our observed DW1 amplitudes is 12 m/s, which is very close to the CTMT result of 9 m/s. The semidiurnal migrating tides, SW2, in the meridional components are stronger from December to October of the next year; SW2 amplitudes in the zonal components are larger from December to May of the next year and peak in July over 95 km. The observed SW2 amplitudes in the zonal components are slightly larger, approximately 8 m/s compared with the 5 m/s CTMT results. Compared with the CTMT results, the SW2 amplitudes based on MRs in the zonal components are stronger from December to April of the next year and peak from June to August over 95 km, which may be due to the lack of radar data in April and July/August. The observed SW2 amplitudes in the meridional components are larger





from February to September, but the altitude of the maximum in February is approximately 10 km lower than the CTMT results, and the weaker amplitudes in October/November are not shown.

Figure 9 also shows the two most significant non-migrating tides, DE3 and SE2, in the CTMT results. The DE3 amplitudes in the meridional components peak in February, September and December, which is consistent with the observations in the left column for the tidal component. However, the DE3 amplitudes in the zonal components are slightly different between the CTMT results and observations. The DE3 amplitudes in the zonal components based on CTMT are larger in January and from April to November, but the observations reach the maximum in January/February and September. Moreover, the CTMT results suggest that the DE3 amplitudes in the meridional components are weaker than those in the zonal components, but the MR observations are the opposite. The observation results of the DE3 amplitudes in the meridional components being stronger than those in zonal components may be because for all three radars, the tidal amplitudes in meridional components are larger (in Figure 4), and other MR observations at low latitudes also support this point, such as the observations by the Kunming MR (Zhao et al., 2012) and the Fuke MR (Jiang et al., 2010). The SE2 tides in the meridional components by CTMT have the largest amplitudes of approximately 4 m/s in March and September; the SE2 tides in the zonal components by CTMT obtain peak amplitudes of approximately 3 m/s in February, June and September, which are somewhat similar to the MR results. The difference between the SE2 tides based on observations and CTMT is that the SE2 amplitudes in the zonal components are stronger in November in the observations, but those in CTMT are weaker; the SE2 amplitudes based on observations in the meridional components reach a maximum in June over 92 km, but the peak amplitudes of the CTMT results in June appear at upper altitudes. Since the zonal and meridional tidal winds of CTMT are based on the TIDI, the different observation principles of the TIDI and MR may cause the same phenomenon to be observed at different altitudes, which leads to those differences. In addition, the DW1 and DE3 tidal components in the zonal components from MRs





reach peak amplitudes in February in the left column of Figure 9, but these peaks do not exist in the right column, which are the CTMT results; in the meridional components, the DW1 and SW2 tidal amplitudes exhibit this phenomenon in February.

Due to uneven temporal sampling of the MR dataset, the fitted tides in February only existed in 2006. A series of studies have reported that the major stratospheric sudden warming (SSW) in 2006 could have an effect on tidal propagation in the MLT region (Hoffmann et al., 2007, Paulino et al., 2011, Sridharan et al., 2009). The SSW event is a major phenomenon in the winter polar region. The significant feature of these events is that over the course of a few days, the westerly stratospheric polar vortex in the winter hemisphere is disrupted, and the mid-stratospheric temperatures increase rapidly by up to 40 K (Andrews et al., 1987). The major SSW event occurs when the stratospheric temperatures increase and the zonal mean circulation at 60° latitude at 10 hPa reverses; the SSW could be classified as a minor event when the temperatures increase, but the zonal mean circulation at a pressure level of 10 hPa does not reverse (McInturff, 1978).

In the research of Butler et al. (2015), the 2006 Northern Hemisphere SSW (NH-SSW) was classified as a major SSW event. Figure 10 presents the DW1, SW2, DE3 and SE2 tidal amplitudes near January 21, 2006, and the dotted lines indicate the SSW central day, which is January 21. In the zonal components, DW1 and DE3 amplitudes exhibit major enhancement after January 21, and the amplitudes peak on approximately February 15; the SE2 amplitudes are also weakly enhanced on January 26. In the meridional components, the DW1, SW2 and DE3 amplitudes exhibit enhancement after the 2006 SSW central day, and the amplitudes also reach a maximum on approximately February 15.

Because of the lack of MR observations in February 2007, the enhancement of tides may not exclude the effects of seasonal variations. To prove that the 2006 NH-SSW event is related to the enhancement of migrating and non-migrating tides after January 21, 2006, Figure 11 shows the residuals of diurnal and semidiurnal tidal amplitudes based on the Kototabang MR, which is the difference between the tidal amplitudes during 2006 SSW and the amplitudes at the corresponding date in the composite year.



A Monte Carlo test was used to evaluate the statistical significance of the residual for the SSW. In this analysis, the time series that follows the uniform distribution was generated randomly. By performing the Monte Carlo procedure 4,000 times, the residual during 2006 NH-SSW was compared with the values from the 4,000 calculations to determine the statistical significance. The areas covered with "circle" symbols denote 90% significance according to the Monte Carlo test, and the areas covered with "cross" symbols denote 95% significance according to the same test. In the zonal components, the enhancement of diurnal and semidiurnal tides is weaker; the diurnal tidal amplitudes increase mainly on February 10, and the semidiurnal tides are enhanced from January 21 to February 2. In the meridional components, the enhancement of both diurnal and semidiurnal tides is very significant; the diurnal and semidiurnal tides are both enhanced from February 10 to February 15 by the 2006 NH-SSW event at over 95% significance. There is a 20- to 25-day delay in the response of the tides to the SSW event, which was reported by Lima et al. (2012). In addition, Eswaraiah et al. (2017) also reported that semidiurnal tides in Antarctica responded to Southern Hemisphere SSW (SH-SSW) after a 35-day delay. Compared with the results of Figure 10, in the zonal components, the minor enhancement of the SE2 amplitudes may be related to the SSW event, but the causal relationship between the DW1/DE3 amplitudes and the SSW event is unclear. According to the enhancement times on January 26 and February 10, the DW1 tide response to the SSW event may exist. In the meridional components, the SW2 amplitudes are clearly modulated by the SSW event, and the enhancement of the DW1 and DE3 tides may also be affected by the 2006 NH-SSW event. Considering the climatological features of DE3 in the meridional components in the CTMT results in Figure 9, the DE3 amplitudes are strongest from January to March, and the enhancement of DE3 amplitudes in the meridional components after February 10, 2006 should be a result of climatological variations. Since the DW1 and DE3 tides are the most dominant components near the Kototabang latitude, the major enhancement of the diurnal tides in the meridional components over Kototabang should mainly be contributed by the DW1 tidal component, which means



that the enhancement of the DW1 tides in the meridional component in Figure 11 is the result of SSW event modulation.

## 5. Summary

Horizontal winds observed by the three MRs in the southern equatorial region are used to investigate the climatology of migrating and non-migrating tides at the mesopause. The neutral wind data were from January 1, 2005, to December 31, 2008, and derived from three MRs, namely, the Cariri MR (7.4° S, 36.5° W), the Kototabang MR (0.2° S, 100.3° E) and the Darwin MR (12.3° S, 130.8° E). According to the tidal fitting simulation based on WACCM winds, the migrating tides could be fitted accurately by the three MRs; the climatology of non-migrating tides could also be well fitted, although the non-migrating tidal amplitudes exhibit differences between the three-point fits and full zonal circle fits.

The migrating and non-migrating tidal amplitudes obtained from these three MRs exhibit significant seasonal and altitudinal variations, and their phases are slightly different between the two wind components. For all tidal components, the monthly mean amplitudes in the meridional components are larger than those in the zonal components. The diurnal eastward-propagating zonal wavenumber-3 tidal amplitudes in the meridional component are the largest, with a peak amplitude of approximately 40 m/s; the largest diurnal migrating tidal amplitudes are approximately 17 m/s, which also occur in the meridional component.

The diurnal migrating tidal amplitudes are strongest in March, reach their second largest amplitudes in October and are weakest in June in both wind components. The semidiurnal migrating tidal amplitudes are stronger from February to October and weaker from September to November. The DE3 tidal amplitudes peak in February, September and December, and the SE2 tidal amplitudes are larger in February and October. These features are very similar to the results from CTMT and are only different in February for nearly all tidal amplitudes in both wind components, which may be the response of tides by 2006 NH-SSW modulation, causing enhancement in February.



*Data availability.* The Kototabang meteor radar data are available from http://database.rish.kyoto-u.ac.jp/arch/iugonet/mwr_ktb/index_mwr_ktb.html. The Darwin meteor radar data are available upon request from Robert. A. Vincent at University of Adelaide (robert.vincent@adelaide.edu.au). The Cariri meteor radar data are available upon request from Paulo P. Batista at National Institute for Space Research (paulo.batista@inpe.br). The CTMT data are available from http://globaldynamics.sites.clemson.edu/articles/ctmt.html.

*Author contributions.* JW and WY designed the study, performed data analysis, prepared the figures and wrote the manuscript. XX initiated the study, and contributed to supervision and interpretation. RAV and IMR provided the Darwin meteor radar data. PPB and RAB provided the Cariri meteor radar data. TT provided the Kototabang meteor radar data. JW and WY contributed equally to this work. WJ are responsible for the WACCM model. TC and XD contributed to interpretation. All authors contributed to discussion and interpretation.

*Competing interests.* The authors declare that they have no conflict of interest.

*Acknowledgments.* This work is supported by the B-type Strategic Priority Program of CAS Grant No. XDB41000000, the National Natural Science Foundation of China (41774158, 41974174, 41831071 and 41904135), the CNSA pre-research Project on Civil Aerospace Technologies No. D020105, the Fundamental Research Funds for the Central Universities (WK2080000125), the China Postdoctoral Science Foundation (2019M652195), the Joint Open Fund of Mengcheng National Geophysical Observatory (MENGO-202008 and MENGO-202009), Key-Area Research and Development Program of the Guangdong Province. (2020B0303020001), and the Open Research Project of Large Research Infrastructures of CAS "Study on the interaction between low/mid-latitude atmosphere and ionosphere based on the Chinese Meridian Project." We acknowledge for the data storage resources from "National Space Science Data Center, National Science & Technology Infrastructure of China. (http://www.nssdc.ac.cn)" We thank Robert. A. Vincent for providing the Darwin meteor radar data, Paulo P. Batista for the data from Cariri meteor radar, and Toshitaka Tsuda for the Kototabang meteor radar data.

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

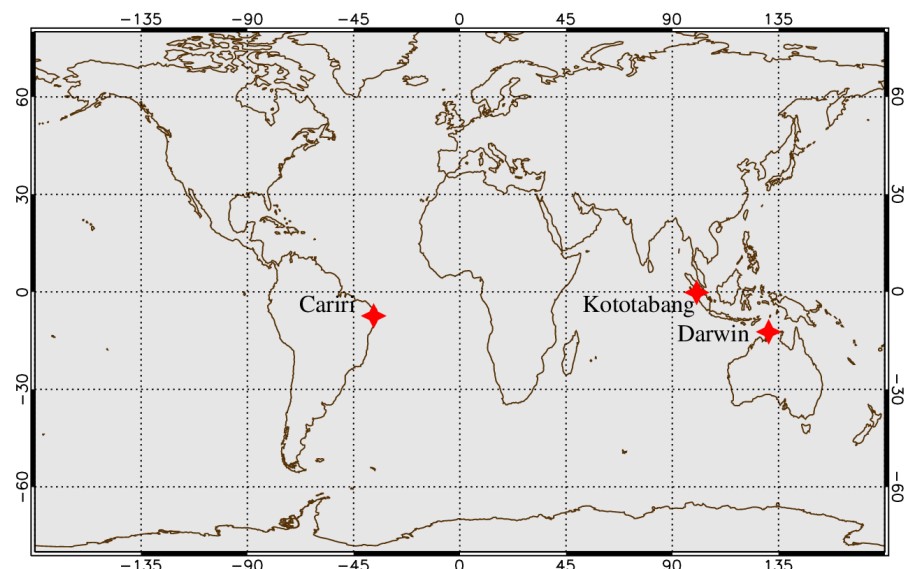

**Figure 1:** The locations of the three MR stations used in the current study.



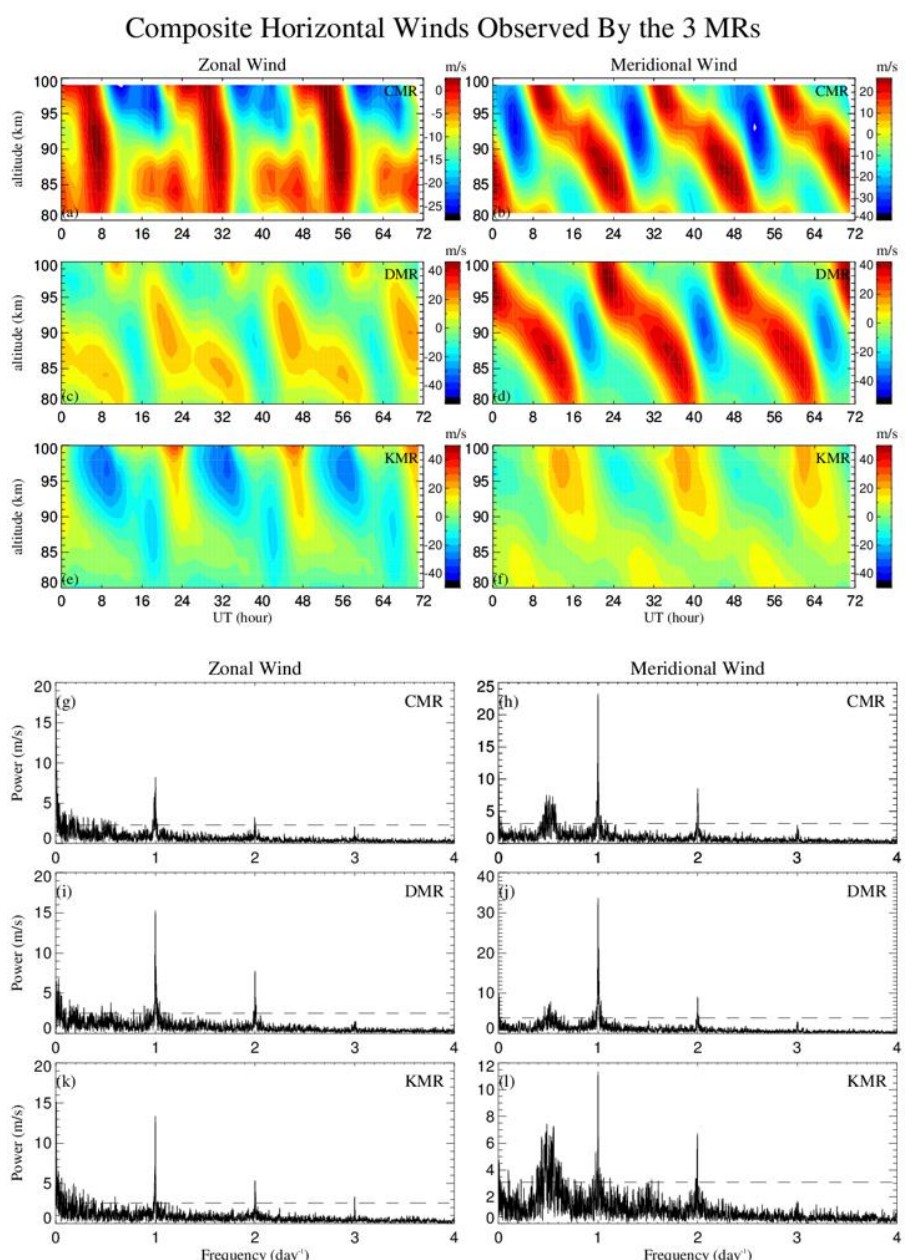

**Figure 2:** (a)-(f) Composite hourly mean zonal and meridional winds observed by the three MRs CMR, DMR and KMR in the altitude range from 80 to 100 km. (g)-(l) Lomb-Scargle analysis for the zonal and meridional winds observed by CMR, DMR and KMR at 90 km.





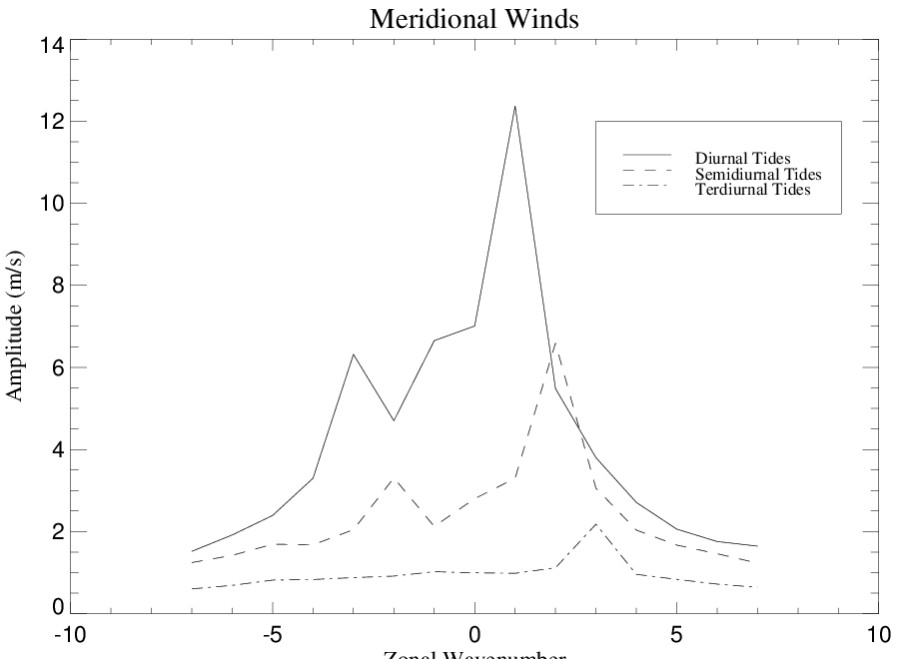

**Figure 3:** All tidal components, including diurnal tides, semidiurnal tides, and terdiurnal tides, averaged at all times and the altitude range from 80 to 100 km by WACCM meridional winds.

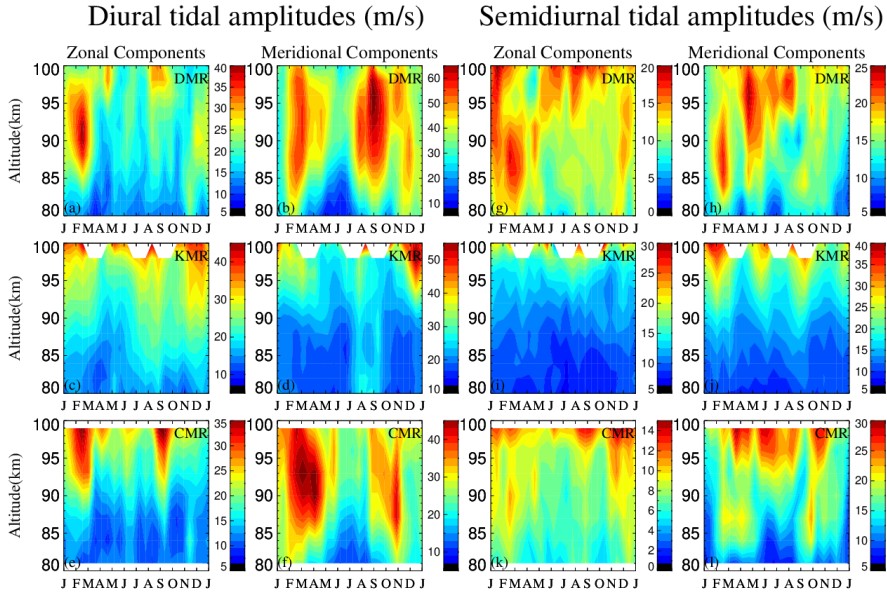

**Figure 4:** Diurnal and semidiurnal tides fitted from DMR, KMR, and CMR during 2005 to 2008 in altitudes ranging from 80 to 100 km in zonal and meridional components.

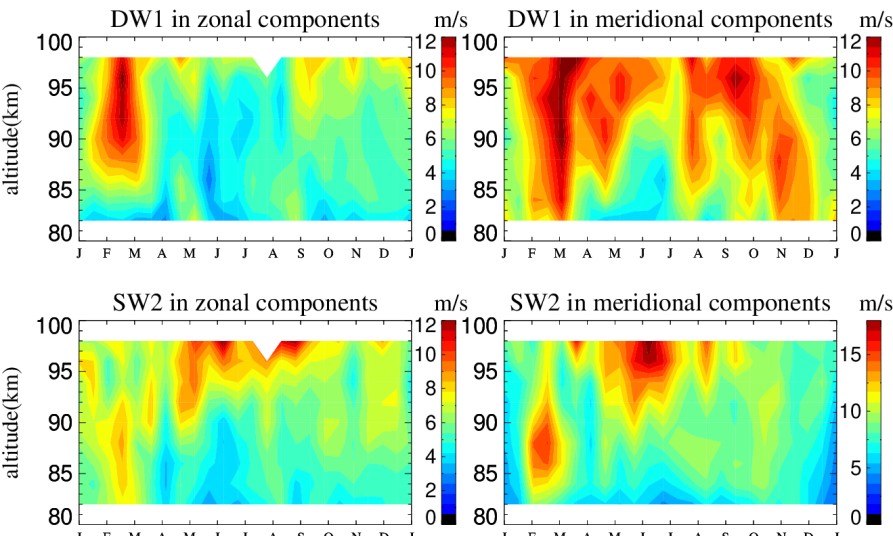

**Figure 5:** Migrating tidal amplitudes, DW1 and SW2, in the zonal and meridional components in composite years at altitudes ranging from 80 to 100 km using the three meteor radars.

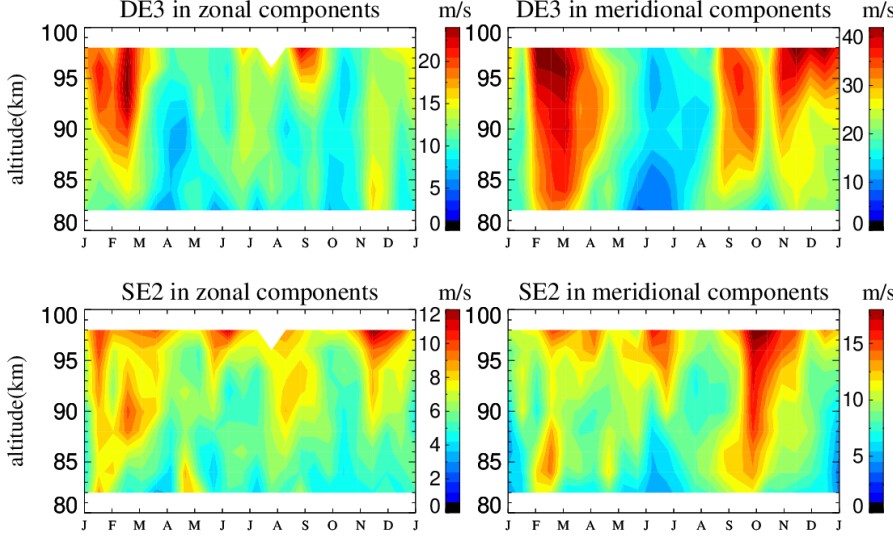

**Figure 6:** Non-migrating amplitudes, DE3 and SE2, in the zonal and meridional components in composite years at altitudes ranging from 80 to 100 km using the three meteor radars.



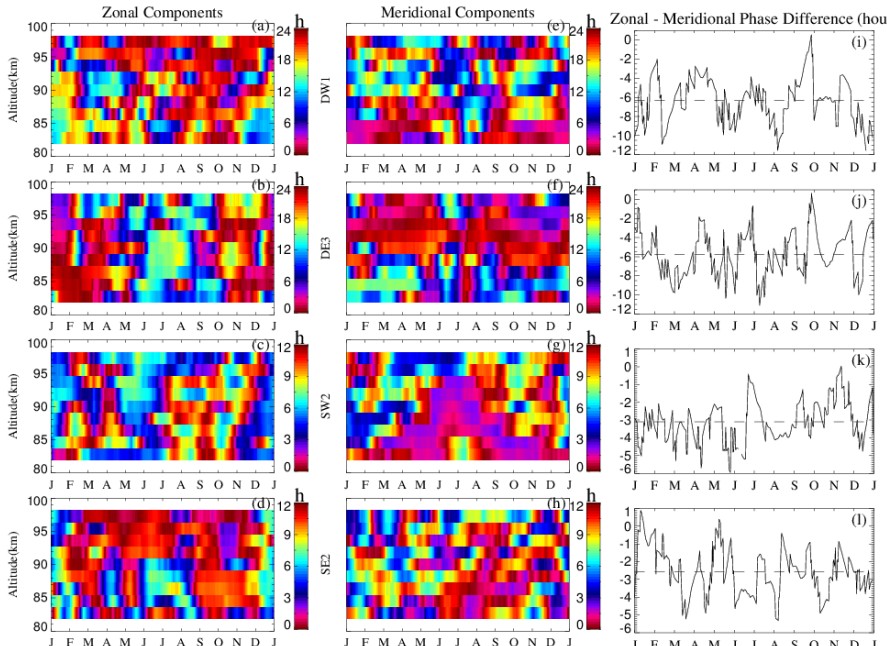

**Figure 7:** (a)-(h) Monthly mean tidal phases in composite years for DW1, DE3, SW2, and SE2 in both wind components at altitudes ranging from 80 to 100 km using the three meteor radars. (i)-(l) Phase Differences between zonal and meridional components averaged over the altitude range of 82 to 98 km for DW1, DE3, SW2, and SE2; The dotted lines indicate the average of phase differences.



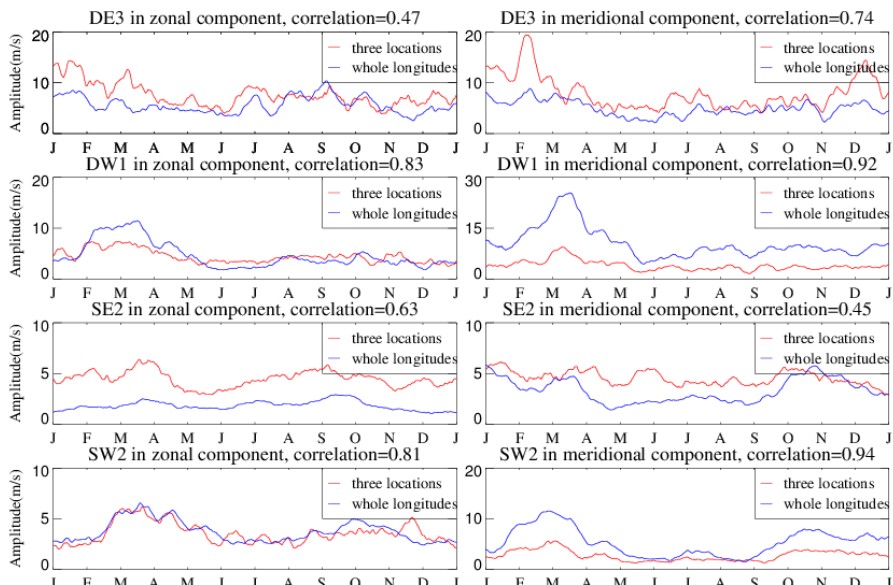

**Figure 8:** Correlation coefficients of the WACCM winds between the three location fits and whole longitude fits for DW1, SW2, DE3 and SE2 at 86 km.



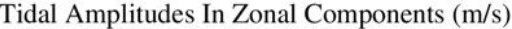

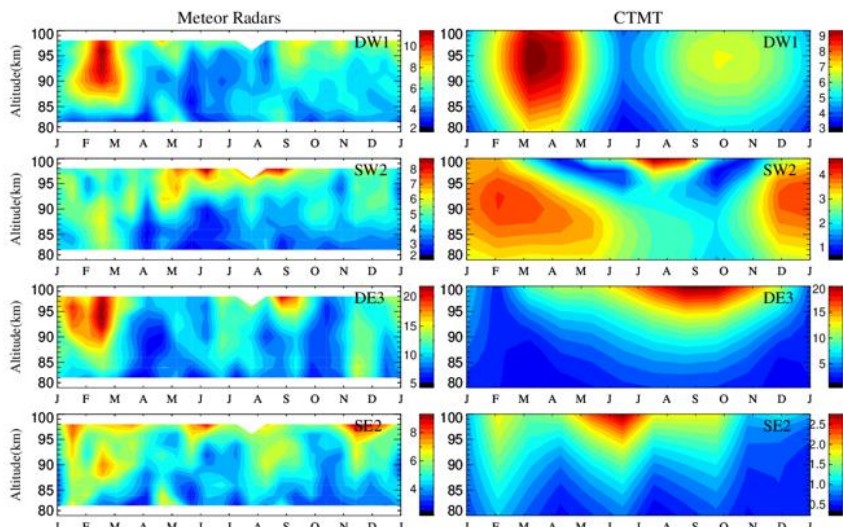

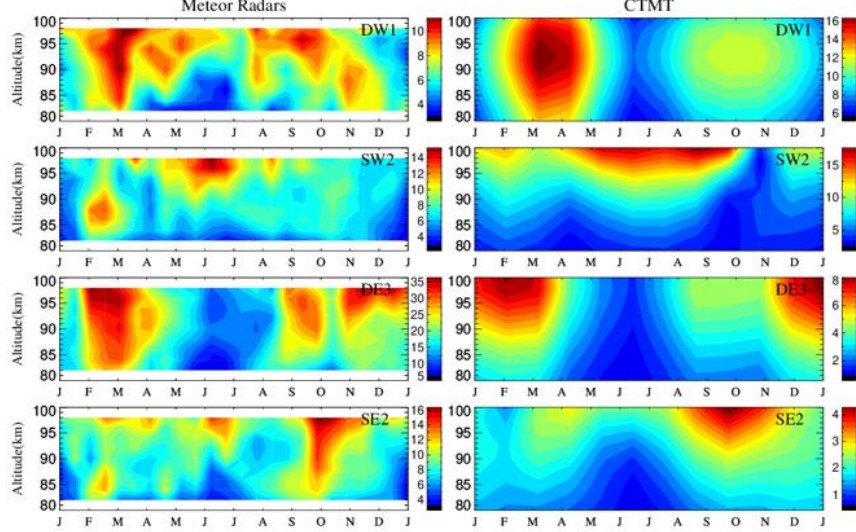

**Figure 9:** MR observations of the DW1, SW2, DE3, and SE2 tidal amplitudes in composite years (in the left column) and the climatologies of the DW1, SW2, DE3, and SE2 tidal amplitudes based on CTMT (in the right column). Different scales for the color bars shows the different maxima of amplitudes for these tidal components.



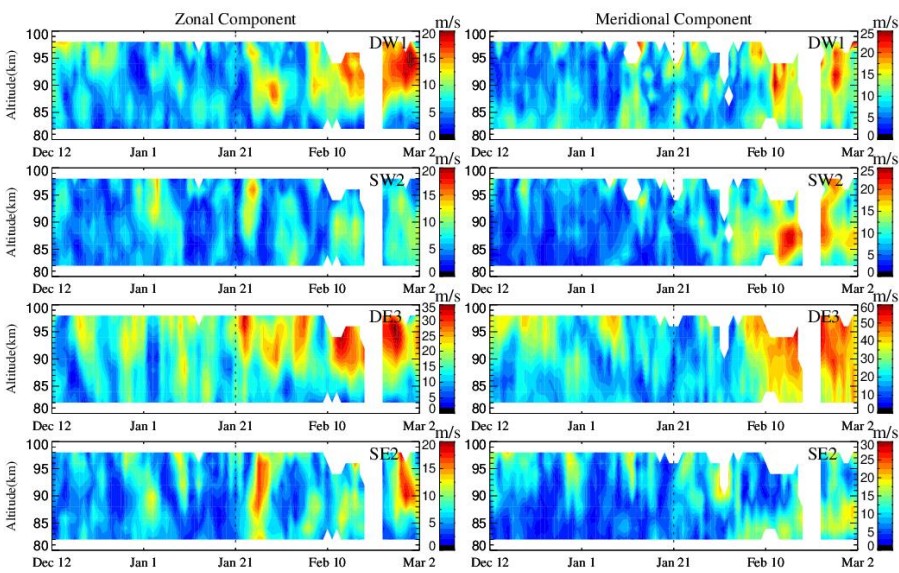

**Figure 10:** The DW1, SW2, DE3 and SE2 tidal amplitudes during 2006 SSW in both wind components at altitudes ranging from 80 to 100 km. The dotted lines indicate January 21.

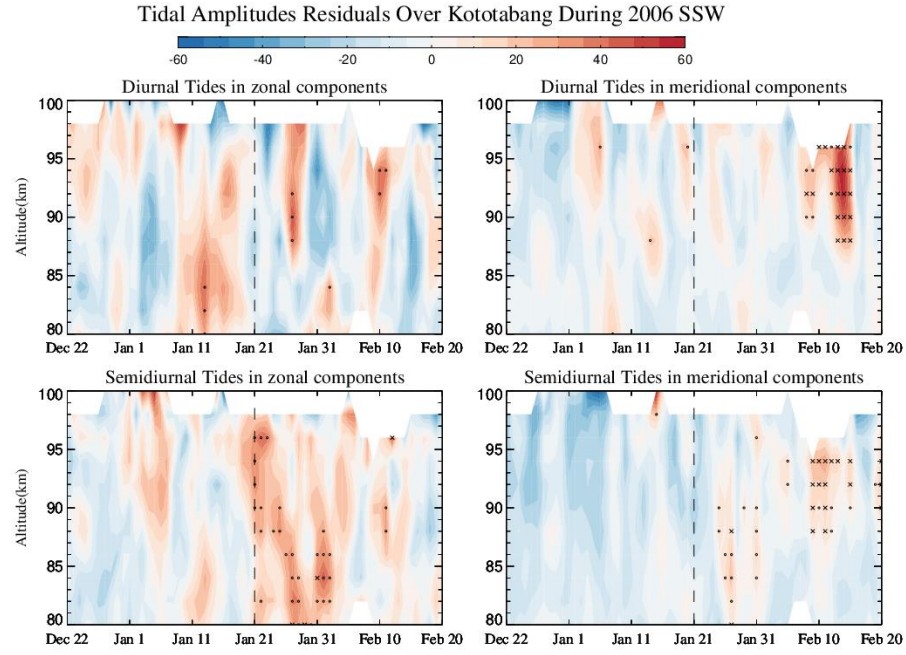

**Figure 11:** Diurnal and semidiurnal tidal amplitude residuals in both wind components





over Kototabang during the 2006 NH-SSW event. The stippling areas indicate the 90% significance according to the Monte Carlo test, and the areas covered with "cross" symbols denote 95% significance according to the same test. The dotted lines also indicate January 21.