# Peer review of "Climatology of migrating and non-migrating tides observed by three meteor radars in the southern equatorial"

_Atmospheric Chemistry and Physics, 2021_

## Author Comment (AC1)

We thank the referees for the very useful suggestions for improving the paper. Our point-by-point responses to the referees' comments are detailed below in blue text, and the changes are shown in the version of the manuscript with track changes.

Response to the Reviewer #1:

Minor comments:

1. page 2, line 13      … could be well fitted by the radars ….

     This sounds strange. I would suggest:      … could be well observed by the radars …

Changed.

2. page 2, line 14      unclear formulation:…. might be overestimated

     overestimated by the model or by the observations?

     what is the reason for the overestimation? The data analysis?

It means that the three-point fitting results of non-migrating tides obtained from WACCM winds are larger than the whole longitudes fitting so the non-migrating tides derived by three meteor radars might be larger than real tides. The unclear formulation has been corrected in revised manuscript.

The reason for the overestimation might be the spatial aliasing that power from high wave numbers can leak into lower ones (Murphy et al., 2006). We have complemented the analysis of using three random longitudinal locations to decomposed tides, which shows that the overestimation of non-migrating tides by three-point fitting exists generally.

Murphy, D. J., Forbes, J. M., Walterscheid, R. L., Hagan, M. E., Avery, S. K., Aso, T., Fraser, G. J., Fritts, D. C., Jarvis, M. J., McDonald, A. J., Riggin, D. M., Tsutsumi, M., and Vincent, R. A.: A climatology of tides in the antarctic mesosphere and lower thermosphere, J. Geophys. Res.-Atmos., 111, 1–17, https://doi.org/10.1029/2005JD006803, 2006.

3. page 2, line 18      unclear:      … slightly different …

     is it again an underestimation of the amplitudes by the model ?

Corrected. It should be the DE3 amplitudes in January-February are slightly different between CTMT results.

4. page 3 , line 13 :     … to fit migrating and …

     I think "fit" is the wrong formulation since the observations are not fitted to the sine waves. The sine waves are fitted to the observations!    Here I would write:

     … to derive migrating and …

Thanks for your suggestion. These formulations have been corrected in the revised manuscript.

5. page 3, line 20      … also demonstrated …

     better:      … also derived ….

Corrected.

6. page 3      Introduction: I am missing 1-2 sentences about the complementarity of tidal measurements from ground and space    Why are ground-based observations necessary?

Thanks for your suggestion. We have complemented a few sentences about why the ground-based observations are necessary in revised manuscript.

7. page 7 line 15      … zonal wavenumber greater than or equal to two cannot be considered …

How can you derive SW2, DE3 and SE2 which have wavenumbers greater equal 2?

This is a formulation mistake, which has been corrected in revised manuscript. It should means that the tides with zonal wavenumber greater than or equal to two cannot be well decomposed with the function (2), while a least square fitting of longitudinal harmonic functions with preassigned zonal wavenumber to observations from different longitude (e.g. Murphy et al., 2006, function (1) in revised manuscript) could well decomposed migrating components and preassigned non-migrating components.

Murphy, D. J., Forbes, J. M., Walterscheid, R. L., Hagan, M. E., Avery, S. K., Aso, T., Fraser, G. J., Fritts, D. C., Jarvis, M. J., McDonald, A. J., Riggin, D. M., Tsutsumi, M., and Vincent, R. A.: A climatology of tides in the antarctic mesosphere and lower thermosphere, J. Geophys. Res.-Atmos., 111, 1–17, https://doi.org/10.1029/2005JD006803, 2006.

Function 1:

$$u(z, \lambda, t) = u_0(z) + c_{1,1}(z)\cos\left(\frac{2\pi}{T}t + \frac{2\pi}{2\pi}\lambda + \varphi_{1,1}(z)\right)$$

$$+ c_{1,-3}(z)\cos\left(\frac{2\pi}{T}t - \frac{2\pi \times 3}{2\pi}\lambda + \varphi_{1,-3}(z)\right)$$

$$+ c_{2,2}(z)\cos\left(\frac{2\pi \times 2}{T}t + \frac{2\pi \times 2}{2\pi}\lambda + \varphi_{2,2}(z)\right)$$

$$+ c_{2,-2}(z)\cos\left(\frac{2\pi \times 2}{T}t - \frac{2\pi \times 2}{2\pi}\lambda + \varphi_{2,-2}(z)\right), \tag{1}$$

where u represents either the zonal or meridional wind; z represents altitude; t represents time; $\lambda$ represents longitude (rad); T equals 24 hours; $u_0$ represents zonal mean zonal wind; $c_{1,1}$, $c_{1,-3}$, $c_{2,2}$, and $c_{2,-2}$ represent the amplitudes of DW1, DE3, SW2, and SE2, respectively; $\varphi_{1,1}$, $\varphi_{1,-3}$, $\varphi_{2,2}$, and $\varphi_{2,-2}$ represent the phases of DW1, DE3, SW2, and SE2, respectively.

Function 2:

$$u(z, \lambda, t) = u_0(z) + \sum_{n=1}^{3}\sum_{s=-7}^{7} c_{n,s}(z) \cos\left(\frac{2\pi n}{T}t + \frac{2\pi s}{2\pi}\lambda + \varphi_{n,s}(z)\right)$$

$$+ \sum_{s=-7}^{7} c_{4,s}(z) \cos\left(\frac{2\pi}{2T}t + \frac{2\pi s}{2\pi}\lambda + \varphi_{4,s}(z)\right), \tag{2}$$

where u represents either the zonal or meridional wind; z represents altitude; t represents time; $\lambda$ represents longitude (rad); n represents temporal wavenumber; s represents zonal wavenumber; T equals 24 hours; $\varphi$ represents phase; $u_0$ represents zonal mean zonal wind; the second section represents tidal components comprising 24-, 12- and 8-hour oscillations; and the third section represents quasi-two-day oscillation components.

8. page 11 line 28    The CTMT is a 2-dimensional model ….
     By the way it seems to be three dimensional    (time, height, latitude)
Corrected.

9. Section 4 is quite long and covering different topics. I would make    2-3 subsections so that the structure of the discussion becomes more visible
Thank you for your suggestion. We have made 3 subsections ("Accuracy of the method used to derive tides", "Comparing the observation results with the CTMT" and "Response of tides to the 2006 SSW event") in the revised manuscript, and the discussion for the accuracy of the method has moved to "Data and Methods".

---

## Author Comment (AC2)

We thank the referees for the very useful suggestions for improving the paper. Our point-by-point responses to the referees' comments are detailed below in blue text, and the changes are shown in the version of the manuscript with track changes.

Response to the Reviewer #2
Specific Comments
The analysis presented here, that attempts to decompose various tidal components from three meteor radar observations is an interesting piece of work. However, the manuscript has several problems that I believe need to be addressed before it is considered for publication. These problems include:
1. The method used in this study for fitting the radar observations over three longitudes to DW1, SW2, DE3 and SE2 tidal components is flawed. As the manuscript states, "the fits of the tides with zonal wavenumbers greater than or equal to two cannot be considered (P7, Lines 15-16)." The fits to DE3 (wavenumber-3) is thus not reliable.

This is a formulation mistake, which has been corrected in revised manuscript. It should mean that the tides with zonal wavenumber greater than or equal to two cannot be well decomposed with the function (2), while a least square fitting of longitudinal harmonic functions with preassigned zonal wavenumber to observations from different longitude (e.g. Murphy et al., 2006, function (1) in revised manuscript) could well decomposed migrating components and preassigned non-migrating components. Murphy, D. J., Forbes, J. M., Walterscheid, R. L., Hagan, M. E., Avery, S. K., Aso, T., Fraser, G. J., Fritts, D. C., Jarvis, M. J., McDonald, A. J., Riggin, D. M., Tsutsumi, M., and Vincent, R. A.: A climatology of tides in the antarctic mesosphere and lower thermosphere, J. Geophys. Res.-Atmos., 111, 1–17, https://doi.org/10.1029/2005JD006803, 2006.
Function 1:

$$u(z, \lambda, t) = u_0(z) + c_{1,1}(z)\cos\left(\frac{2\pi}{T}t + \frac{2\pi}{2\pi}\lambda + \varphi_{1,1}(z)\right)$$

$$+ c_{1,-3}(z)\cos\left(\frac{2\pi}{T}t - \frac{2\pi \times 3}{2\pi}\lambda + \varphi_{1,-3}(z)\right)$$

$$+ c_{2,2}(z)\cos\left(\frac{2\pi \times 2}{T}t + \frac{2\pi \times 2}{2\pi}\lambda + \varphi_{2,2}(z)\right)$$

$$+ c_{2,-2}(z)\cos\left(\frac{2\pi\times2}{T}t - \frac{2\pi\times2}{2\pi}\lambda + \varphi_{2,-2}(z)\right), \tag{1}$$

where u represents either the zonal or meridional wind; z represents altitude; t represents time; $\lambda$ represents longitude (rad); T equals 24 hours; $u_0$ represents zonal mean zonal wind; $c_{1,1}$, $c_{1,-3}$, $c_{2,2}$, and $c_{2,-2}$ represent the amplitudes of DW1, DE3,

SW2, and SE2, respectively; $\varphi_{1,1}$, $\varphi_{1,-3}$, $\varphi_{2,2}$, and $\varphi_{2,-2}$ represent the phases of DW1, DE3, SW2, and SE2, respectively.
Function 2:

$$u(z, \lambda, t) = u_0(z) + \sum_{n=1}^{3} \sum_{s=-7}^{7} c_{n,s}(z) \cos\left(\frac{2\pi n}{T} t + \frac{2\pi s}{2\pi} \lambda + \varphi_{n,s}(z)\right)$$

$$+ \sum_{s=-7}^{7} c_{4,s}(z) \cos\left(\frac{2\pi}{2T} t + \frac{2\pi s}{2\pi} \lambda + \varphi_{4,s}(z)\right), \tag{2}$$

where u represents either the zonal or meridional wind; z represents altitude; t represents time; $\lambda$ represents longitude (rad); n represents temporal wavenumber; s represents zonal wavenumber; T equals 24 hours; $\varphi$ represents phase; $u_0$ represents zonal mean zonal wind; the second section represents tidal components comprising 24-, 12- and 8-hour oscillations; and the third section represents quasi-two-day oscillation components.

2. The model results (shown in Figure 3) show large amplitudes for D0, DE1 and DE2 components in addition to DW1 and DE3. Specifically, D0 and DE1 are shown to have larger amplitudes than DE3 and both are stronger than semidiurnal components. However, the authors fit the data to only DW1, SW2, DE3 and SE2 (stated in P7, lines 16-17) and other components are not included. Also, DE2 tides can approach large amplitudes as demonstrated in previous climatological studies (e.g. Forbes et al., 2008), thus should be included.

Forbes, J. M., X. Zhang, S. Palo, J. Russell, C. J. Mertens, and M. Mlynczak (2008), Tidal variability in the ionospheric dynamo region, J. Geophys. Res., 113, A02310, doi:10.1029/2007JA012737.

Thanks for this suggestion. In revised manuscript, the model to infer tidal components that dominate in the meteor radar observations has been replaced with CTMT, which are derived with SABER and TIDI, and the result has been shown in Fig 4 (Fig 3 in original manuscript) that the dominant diurnal non-migrating components are DE3 components and the dominate semidiurnal non-migrating components are SE2. SE2 is the largest non-migrating components in semidiurnal tides, we decomposed this SE2 component to discuss the semidiurnal non-migrating component; in fact, we have added the components which stronger than SE2, such as DE2, DE1 and D0, to function (1) to decomposed tides, but the accuracy is worse than the results by function (1) which only decompose four tidal components.

DE2 components are slightly weaker than DE3 in MLT region by CTMT modelling although that component can reach large amplitudes in SABER temperature. We have added a few sentences to discuss the DE2 component; however, by the reason that we only want to discuss the largest non-migrating diurnal component derived by meteor radars, DE2 should not be included in present study. The DE2 components obtained from ground-based observations will be reported in a following paper.

3. This work uses the model results to infer tidal components that dominate in the meteor radar observations, but no validation of the model is provided or referenced. The modeled tides should be compared with other observational data for the same time periods as the radar data. In addition, the tidal amplitudes have a large seasonal variation, so the model-observation comparisons should be conducted for individual seasons. These model validations are lacking in the manuscript.

In revised manuscript, the model to infer dominant tidal components has been replaced with CTMT, which is derived from TIDI and SABER, and we added references about validation of the model (Oberheide et al., 2011); also, it shows that the dominant diurnal non-migrating components are DE3 components and the dominate semidiurnal non-migrating components are SE2 in Fig 4.

Furthermore, we present the composite WACCM winds in Figure 2 (m) – (r), which are similar to the MR observations.

Oberheide, J., Forbes, J. M., Zhang, X., and Bruinsma, S. L.: Climatology of upward propagating diurnal and semidiurnal tides in the thermosphere. Journal of Geophysical Research, 116, A11306, https://doi.org/10.1029/2011JA016784, 2011.

[Figure]

**Figure 2:** (a)-(f) Composite hourly mean zonal and meridional winds observed by the three MRs, CMR, DMR and KMR, in the altitude range from 82 to 98 km averaged during the time period from January 30, 2006, to March 1, 2006. Black dotted lines in Figure 2 (a)-(f) indicate the zero-wind lines; black dashed lines in Figure 2 (g)-(l) indicate the minimum phase of diurnal oscillations at each altitude. (g)-(l) Hourly mean diurnal and semidiurnal winds reconstructed from all extracted tidal components at the three MR locations averaged during the same time period. (m)-(r) Composite WACCM zonal and meridional winds at the three MR locations during the same time period.

[Figure]

**Figure 4:** Diurnal and semidiurnal tides with different zonal wavenumbers from eastward-propagating zonal wavenumber 4 (-4) to westward-propagating zonal wavenumber 3 (+3) averaged at all times and altitudes ranging from 82 to 98 km by CTMT models. The green squares indicate the diurnal components, and the red triangles indicate the semidiurnal components. The black crosses mark the dominant migrating and non-migrating components in diurnal and semidiurnal components, respectively.